**Data Availability Statement:** The data and materials are available on the Open Science Framework: https://osf.io/ezqfp/.

**Funding:** The author(s) received no specific funding for this work.

# Corporate social responsibility and corporate financialization—Based on information effect and reputation insurance effect

**Lei Lei** [1]*, **Di Zheng**[2], **XuDong Chen**[1]

**1** School of Accounting, Southwestern University of Finance and Economics, Chengdu, Sichuan, China,
**2** Institute of Regional Economics, Sichuan Academy of Social Sciences, Chengdu, Sichuan, China

* Leileio@smail.swufe.edu.cn

## Abstract

China has witnessed the trend of corporate financialization (CF) with some potential risks as the economy slows down its pace in the past decade. In this paper, we explore whether corporate social responsibility (CSR) could work as an information channel to restrain CF or as reputation insurance to promote CF. We find a significant positive relation between CSR and CF, especially for non-SOEs and enterprises with low ownership concentration and high CSR scores. It verifies that the reputation insurance effect by CSR outweighs the information effect and denies the opposite. The results prove robust in tests including sensitivity and endogeneity test. By expanding the scale and adding new aspects to the discussion about how CSR affects CF, this paper provides valuable empirical support to both theorists and practitioners.

## Introduction

In the 1980s, rapid development in the marketing economy increased capital's profitability and transformed its creative value into the basic economic driving force. Thereafter, Internet finance further intensified financial expansion. Nowadays, financial capital is dominating with the weakening of real economy, as shown by the fact that the financial output in the U.S. is increasing yearly and has surpassed that of manufacturing output. The average share of financial output in GDP was 17.1% in 1965-1980, while the share rose dramatically to 28.9% in 2000-2015. In contrast, the share of manufacturing output in GDP declined significantly: from an average of 49.1% in 1965-1980 to 20.9% in 2000-2015. The above data show that financial capital is gradually replacing industrial capital as the new force dominating the U.S. economy, and corporate financialization (CF) in developed economies is irreversible.

In recent years, as a representative of countries in economic transition, China has witness both hampered real economy and the trend of corporate financialization. On the one hand, capital has gradually defied its original goal to serve the real economy and shifted to the internal circulation in the financial system. On the other hand, an increasing number of non-financial corporates are detached from their original main business and rely heavily on investment profits in the virtual economy. CF has gradually evolved into an important strategy for

**Competing interests:** The authors have declared that no competing interests exist.

**Fig 1. Scale of financial assets and ratio of financial assets to total assets of non-financial listed companies.**

non-financial enterprises to improve financial condition and pursue short-term profits. Fig 1 shows the average value of the financial assets scale and the ratio of financial assets to total assets for China's non-financial listed companies from 2008 to 2019. It is not difficult to see that the amount of financial assets and the average ratio of financial assets showed a rapid growth trend from 2011 to 2018, which is believed to be financialization [1–4]. Indeed, financialization can quickly obtain liquidity to reduce the cost of financial distress and thus promote the industry development [5, 6], but excessive financialization will lead to the draining of investment in the real economy, which will eventually lead to the unbalanced development between the over-heated virtual economy and the under-developed real economy with consequent harms to the public interest.

In the 1970s, large enterprises strengthened their voice in the society with expansion all around the world, turning CSR a widely concerned topic. Subsequently, the CSR movement emerged and flourished in developed economies. With the advancement of globalization, large multinational enterprises gradually relocated their supply and production chains to developing countries and introduced CSR awareness and requirements there as a result. Facing the global impact of CSR concepts from the international community, Chinese enterprises have been increasing awareness and enhancing behaviors in CSR. With the parallel advancement of theoretical research and corporate practice, CSR practices have gradually stepped into the stage of standardized development in China. However, according to the statistics of Runling Global Responsibility Rating (RKS), there are still a series of problems in the fulfillment of social responsibility by Chinese enterprises: First, although the number of enterprises disclosing social responsibility reports has increased, the overall proportion is low. In 2019, for example, a total of 942 listed companies disclosed social responsibility reports, but they only accounted for 24.94% of the overall market. Secondly, the motivation of enterprises to voluntarily disclose social responsibility information is not strong. only 57.32% of the enterprises that released social responsibility reports in 2019 were voluntary, and the rest were required to disclose. Finally, the overall quality of CSR reports is low, and many enterprises' CSR reports are called "cloned social responsibility reports", in which all expressions are copied except for changes in specific events and figures. In addition, corporate hypocrisy, political rent-seeking and other CSR alienation are particularly prominent in China's CSR practices. The profit-seeking nature

of enterprises will eventually offset or even distort their efforts in social responsibility, and lead to serious economic consequences. While the existing literature reveals that CSR may have adverse effects such as the risk of stock price collapse, not enough attention has been paid to the impact of CSR on corporate investment behavior, especially on the investment behavior of financial assets.

According to the theory of social exchange, CSR can help enterprises to enhance reputation and obtain strategic resources such as capital, which in turn affects the investment preference. Existing documents indicate that CSR helps to reduce financing costs, obtain tax relief, and acquire financial subsidies, which has a positive impact on financial performance [7, 8]. On the contrary, there are also research claim that CSR is a tool to cover up corporate misconduct, which provides a shield for management's self-interest behavior and increases the stock price crash risk [9, 10]. So, is CSR a reputation insurance tool to cover up negative information or an information transmission tool to promote its development? Specifically, when it comes to the impact on corporate financial investment decisions, does CSR promote financialization or inhibit financialization?

To answer the above questions, this paper investigates the relationship between CSR and CF. Most relevant to this subject is the research of Liu et al. (2019) and Gu et al. (2020) [11, 12]. Among them, Liu et al. (2019) found CSR as a management tool to inhibit CF based on the instrumental hypothesis [11]; Gu et al. (2020) found that CSR promotes CF and verified that the motivation of CF is "investment substitution" rather than "reservoir" based on the shareholder value hypothesis [12]. In summary, this paper further discusses two hypotheses of CSR: the instrument hypothesis and the shareholder value hypothesis and finds that both influence CF through the reputation insurance effect and the information effect. The information effect is that CSR reports provide stakeholders with access to "soft information" and strengthen their ability to monitor corporate governance, which can effectively inhibit short-term profit-seeking behavior and reduce the degree of CF. The reputation insurance effect is reflected when companies promote charitable donations and environmental management, which enhances the public's goodwill toward corporates and tolerance of misconduct. Facing with weakened cash flow and the need to obtain short-term income, the corporates will enhance financialization behavior for the motive of "investment substitution". Taking these into consideration, using a sample of Chinese non-financial listed companies from 2009-2019, this paper examines the relationship between CSR and CF and tests the impact mechanisms.

Compared with the previous literature, the main contributions of this paper are reflected in the following three points: First, the existing research on the relationship between CSR and CF has inconsistent conclusions. This paper provides an important supplementary document in this field, as well as empirical evidence for countries with transformed economies. Second, this paper explores how CSR affects CF, which provides potent evidence to understand CSR better. Thirdly, this paper focuses on the purpose to fulfill CSR, and discusses the economic consequences of CSR in the practical field from the perspectives of management and shareholders.

## Literature review and research assumptions

### Literature review of corporate financialization

Corporate financialization (CF) refers to an increase in the participation of the real economic sector in financial investment activities and an increase in the share of financial gains in profits [1, 3]. With reference to Cai and Ren (2014) and Wang et al. (2017), this paper defines CF as "the trend that non-financial enterprises reduce investment in the real economy and increase investment in financial assets (including commodities with investment properties such as real estate)" [13, 14].

There are mainly "reservoir" and "investment substitution" theories to explore the motivation of CF. The "reservoir" theory holds that the purpose of holding financial assets is to reserve liquidity to prevent the risk of capital chain fracture caused by cash flow shocks [5, 6]. However, the "investment substitution" theory holds that the purpose of CF is to maximize profits. When the return rate of financial investment is higher than that of real economy investment, enterprises will replace real economy investment with financial assets investment [15, 16]. Hu et al. (2017) found that the CF in China is mainly in line with the "investment substitution" theory, i.e., the pursuit of a high rate of return on financial investment is the main motivation for Chinese corporate financialization [17].

## Literature review of corporate social responsibility

Corporate Social Responsibility (CSR) is generally regarded as "actions and strategies of an enterprise to consider the expectations of stakeholders and maintain the triple bottom line of economic, social and environmental benefits under specific circumstances" [18]. What impact will CSR have on corporate financial behavior? Scholars have launched a fierce and lasting debate on this issue. At present, it can be concluded into two viewpoints: the shareholder value hypothesis which is beneficial to improving corporate value and the management self-interest hypothesis which is harmful to corporate value [10, 19].

**1. Shareholder value hypothesis.**   Shareholder value doctrine holds that CSR is essentially an intangible asset that maintains the foundation of a relationship. In the long run, fulfilling CSR will benefit the enterprise from three aspects including the system, organization and individual. From the institutional level, CSR can improve corporate reputation, shape corporate citizenship image, and enhance institutional legitimacy [20]. For example, Macao's gaming companies use CSR as a strategic response to address the legitimacy gap, to obtain support from the government and public [21]. In addition, tainted enterprises can repair their reputation through donation in 2008 Wenchuan earthquake [22]. At the organizational level, firstly, the performance of CSR helps enterprises to obtain business benefits, including improvement in financial performance, access to commercial credit, reduction of financing costs and gaining political resources [20, 23–28]. Secondly, CSR performance can also help enterprises improve business practices such as improving the quality of financial reporting, enhancing customer satisfaction [29, 30], etc. At the individual level, CSR contributes to the improvement of employees' emotional identity, work performance and citizenship behavior [31–34].

**2. Management self-interest hypothesis.**   Principal-agent theory is the theoretical basis of the management self-interest hypothesis in CSR. Based on this theory, scholars represented by Friedman believe that managers are not efficient and reliable social responsibility bearers. Enterprises undertake CSR in order to maximize the managers' own interests, but not to maximize the stakeholders'. Some researchers believe that CSR increases the unnecessary costs and operating risks and puts the company in a disadvantageous position in the competition [35, 36]. Friedman denounced CSR as "a kind of fundamental destructive doctrine in a free society", insisting that the only social responsibility of enterprises is "to earn as much money as possible on the premise of abiding by laws and corresponding moral standards" [37, 38]. Simultaneously, the opponents criticized the content of CSR is vague, the object of obligation is general and unclear [39].

## Hypothesis development

As a reflection of the interaction between enterprises and stakeholders, CSR not only conveys more internal information to the outside, but also affects the reputation image of the enterprise in the market. Based on the two doctrines of CSR (shareholder value VS management self-

interest), this paper puts forward two opposite hypotheses: information effect and reputation insurance effect.

**1. Information effect.** It has been pointed out that better CSR reflects higher cultural and ethical standards of enterprises, lower motivation to conceal bad news and better information transparency [40]. CSR reports help to alleviate agency problems that may be caused by insiders' information advantages, such as insider trading, tax transfer and earning manipulation, etc., which have a certain inhibitory effect on management's selective information disclosure and negative information management [41–43]. From the perspective of investors, as CSR conveys more non-financial information, external investors can better understand the enterprises' financial condition, so as to implement more reasonable and wise investment decisions, make the enterprises operation information reflected in the stock price more authentic and effective, and avoid phenomena such as inflated stock price and bubbles [44].

Based on the "information effect", CSR, as an important non-financial information of an enterprises, can reflect the operating efficiency, risk management, innovation, and other aspects of the enterprises, which is conducive to reducing the information asymmetry between stakeholders and the enterprises and forming a supervisory and incentive role for the management. Kim et al. (2014) confirmed that as CSR increases the information transparency of enterprises and conveys more true information to the market, it is beneficial to restrain managers' motivation to hide bad news and reduce stock price crash risk [45]. In short, CSR can improve the quality of information transmission between the enterprises and stakeholders. Under the joint supervision of stakeholders, the internal agency problem will be effectively alleviated. Naturally, the investment decision-making of an enterprise will aim at promoting the long-term development and safeguarding the interests of stakeholders, thus focusing on the main business and reducing the irrational profit-seeking of financial assets. Accordingly, we propose hypothesis H1a:

H1a: If the information effect is true, CSR will inhibit CF.

**2. Reputation insurance effect.** CSR is targeted at various stakeholders, such as consumer demand, product quality, staff development and environmental performance. They send signals to stakeholders that enterprises are not completely self-interested, which is beneficial to establishing a good citizenship image and forming reputation capital. When the enterprises face threat of negative events, CSR can play a role like insurance which can mitigate negative effects [46]. Even if the enterprise breaks out negative news itself, investors may attribute it to the clumsiness of managers rather than malice [47, 48]. Under the condition of uncertain motivation, external market will assume the right of the enterprises to be innocent (i.e., believe that the management based on good motivation but produced bad results), thus reducing the degree of "sanction" (i.e., mass selling of shares) [46].

Based on the "reputation insurance effect", instead of transmitting effective information to investors, CSR becomes a tool to disguise managers' profit-seeking behavior. Friedman believes that CSR is a waste of shareholders' wealth. As an agent of shareholders, managers may use enterprises resources to seek their own interests [37]. Godfrey proposed that enterprises accumulate reputation capital by fulfilling CSR, to mitigate the negative evaluation of investors and play a role of reputation insurance when enterprises encounter negative events [46, 47]. Previous studies have shown that the reputation insurance function of CSR will increase earning management and stock price crash risk. Therefore, under the cover of the good image created by CSR, managers may abandon long-term industrial investment and pursue short-term economic benefits through investment in financial assets, which leads to an increase in CF. Therefore, we propose hypothesis H1b:

H1b: If the reputation insurance effect is true, CSR will promote CF.

The information effect and reputation insurance effect of CSR show opposite consequences among enterprises with different equity nature. If the information effect is true, the inhibitory effect of CSR on CF is more obvious in non-SOEs. Compared with the private enterprises, SOEs undertake more social functions and pay more attention to balance social responsibilities and their own interests. Therefore, the inhibition of CSR on financialization is more significant in non-SOEs. If the reputation insurance effect is true, the promotion effect of CSR on financialization is more obvious in non-SOEs. Because non-SOEs suffer from stress about capital and uncertain profits, they are more motivated to participate in financial speculation to obtain more profits. Therefore, the role of CSR in promoting CF is more significant in non-SOEs. Based on the above analysis, the following hypotheses are made:

H2a: If the information effect is true, the negative relationship between CSR and CF is more significant in non-SOEs.

H2b: If the reputation insurance effect is true, the positive relationship between CSR and CF is more significant in non-SOEs.

The information effect and reputation insurance effect of CSR are different in diverse ownership concentration enterprises. If the information effect is true, the external supervision of enterprises with low ownership concentration will be greatly strengthened, and their financialization speculation will also be significantly inhibited. If the reputation insurance effect is true, enterprises with low concentration of equity will participate in financial speculation more strongly, and their CF is more significant. As for the enterprises with relatively low ownership concentration, they have weak external supervision and weak internal governance, and the self-interest of the management is more significant. Therefore, they have stronger motivation and ability to participate in financial speculation and seek short-term benefits. Accordingly, we propose the following hypotheses:

H3a: If the information effect is true, the negative relationship between CSR and CF is more significant in low ownership concentration enterprises.

H3b: If the reputation insurance effect is true, the positive relationship between CSR and CF is more significant in low ownership concentration enterprises.

The information effect and reputation insurance effect of CSR are different among enterprises with different social responsibility performance. When the reputation insurance effect is true, the better CSR performance, the better it will help establish a good public image in the external market and form better reputation insurance. Quan and Xiao (2016) found that CSR is more "opportunity-driven" than "value-driven" in China [49]. Such self-interest behavior cannot reflect the moral awareness of enterprises. Currently, enterprises with high CSR score are more likely to conceal their financial speculation, and the cost of financial investment is lower. When the information effect is true, enterprises with high CSR score will be more likely to inhibit CF. Because enterprises with high CSR pay more attention to the relationship with stakeholders, they will independently improve the rights and value of stakeholders, consciously regulate the operation behavior, and seek long-term sustainable development, thus they are more likely to give up short-term financial speculation [50]. Therefore, the following hypotheses are further made:

H4a: If the information effect is true, the negative relationship between CSR and CF is more significant in high CSR score enterprises.

H4b: If the reputation insurance effect is true, the positive relationship between CSR and CF is
   more significant in high CSR score enterprises.

## Research design

### Sample selection and data sources

This paper selects the listed companies in China from 2009 to 2019 as the research samples.
They were processed as follows: (1) exclude the financial industry and real estate industry sam-
ples; (2) remove samples in lack of financial and corporate governance data. To eliminate the
influence of extreme values, we winsorize all continuous variables from 1% to 99% level. Even-
tually, we obtain 20253 sample observations. The CSR data are derived from the corporate
social responsibility scoring index on Hexun.com. The financial data and corporate gover-
nance data are derived from the RESSET and CSMAR database.

### Measurement of variables

**1. Corporate Financialization (CF).**   It is generally believed that corporate financial assets
mainly include 12 balance sheet items such as transactional financial assets, derivative financial
assets, other receivables, buy-back financial assets, non-current assets due within one-year,
other current assets, loans and advances, available-for-sale financial assets, held-to-maturity
investments, long-term equity investments, investment real estate and other non-current
assets [12, 51, 52]. Among them, the "other receivables" account can reflect the scale of corpo-
rates' shadow banking activities based on broad commercial credit and private lending. Long-
term equity investment in the financial industry is a typical financial speculation activity, indi-
cating that corporates indirectly engage in shadow banking activities through holding financial
subsidiaries, associates, or joint financial corporates. As the real estate industry is showing typ-
ical virtualization characteristics, this paper defines real estate investment as financial assets.
Based on this, this paper obtains the above 12 balance sheet account data from the financial
statements and sums up the data to obtain the financial assets held by the corporate. Therefore,
we use the relative value and absolute value of financial assets to measure the degree of corpo-
rate financialization, where CF1 is the ratio of total financial assets to total assets and CF2 is
the natural logarithm of total financial assets [17, 52].

**2. Corporate Social Responsibility (CSR).**   This paper selects the CSR scoring index of
listed companies by Hexun.com to measure CSR performance. The evaluation system starts
from five aspects: shareholder responsibility, employee responsibility, consumer responsibility,
environmental responsibility, and social responsibility, involving 13 secondary indicators and
37 tertiary indicators, and adjusts the weight proportion of the five primary indicators accord-
ing to the industry. This paper uses the social responsibility score divided by 100 as an indica-
tor to measure CSR performance. The higher the score, the better CSR performance. We also
use the CSR report disclosures of Runling Global for robustness testing. If the listed company
discloses the CSR report in the current year, the CSR virtual variable is assigned to 1, otherwise
it is assigned to 0.

**3. Control variable.**   The selection of control variables is based on the previous research
[12, 51, 52]. (1) Financial investment yield is calculated as the sum of investment income,
profit on changes in fair value and net exchange gain, net of investment income to associates
and joint ventures, divided by the total financial assets [51, 53]. Operating investment yield is
calculated as operating income minus operating costs, taxes, period expenses and impairment
losses on assets, divided by the total operating assets. (2) Corporate financial variables,

**Table 1. Definition of variables.**

| Variable name | Variable Symbol | Variable Definition |
|---|---|---|
| Corporate Financialization | CF1 | Total financial assets held by enterprises / total assets |
| | CF2 | Natural logarithm of total financial assets held by enterprises |
| Corporate Social Responsibility | CSR | The comprehensive CSR rating of listed companies published by Hexun.com. |
| Financial investment yield | Fiy | Financial investment income / total financial assets |
| Operating investment yield | Oiy | Operating investment income / total operating assets |
| Scale | Size | Natural logarithm of total assets |
| Ownership nature | SOE | 1 = state-owned enterprises, 0 = non-state-owned enterprises |
| Leverage ratio | Lev | Total liabilities / total assets |
| Growth | Grow | Operating income growth rate |
| Profitability | ROA | Net profit / average total assets |
| Cash holdings | Cash | Cash and equivalents / total assets |
| Book-to-market ratio | MB | Total assets / total market value |
| Management fee rate | Mfr | Management expenses / revenue from main business |
| Finance charge rate | Fcr | Finance expenses / revenue from main business |
| Tax | Tax | Income tax expense / operating income |
| Equity concentration | Econ | The largest shareholder's shareholding ratio |
| Equity restriction | Eres | The sum of the shareholding ratios of the 2nd largest shareholder to the 10th largest shareholder |
| Management shareholding | Ms | Number of management shares / total shares |
| Institutional investors' shareholding | Is | Number of shares held by institutional investors / total shares |
| Proportion of independent directors | Ind | Proportion of independent directors on the board |
| Duality | Dual | 1 = CEO duality; 0 = CEO non-duality |

including size, ownership nature, leverage ratio, growth, profitability, book-to-market ratio, management expenses, finance expenses and tax. (3) Corporate governance variables, including equity concentration, equity restriction, management shareholding, institutional investor shareholding, independent director ratio, and duality. The specific definitions of variables are shown in Table 1.

## Regression model

This paper verifies the relationship between CSR and CF and the basic regression model is set as follows. The model controls industry fixed effect (*Industry*), province fixed effect (*Province*) and annual fixed effect (*Year*).

$$CF_{i,t} = \beta_0 + \beta_1 CSR_{i,t} + \text{Controls}_{i,t} + \sum \text{Industry} + \sum \text{Province} + \sum \text{Year} + \varepsilon_{i,t} \qquad (1)$$

## Empirical results and analysis

### Descriptive statistics

Table 2 presents descriptive statistical results for the key variables. The average value of the relative scale of financialization (CF1) is 0.120, indicating that the average value of the proportion of financial assets to the total assets is 12%; The maximum value is 0.618 and the minimum is

**Table 2. Descriptive statistical analysis.**

| Variables | Obs | Mean | Std. | Min | p25 | Med | p75 | Max |
|---|---|---|---|---|---|---|---|---|
| CF1 | 20253 | 0.120 | 0.123 | 0.001 | 0.033 | 0.079 | 0.163 | 0.618 |
| CF2 | 20253 | 19.26 | 1.834 | 14.23 | 18.18 | 19.38 | 20.49 | 23.49 |
| CSR | 20253 | 0.240 | 0.149 | -0.029 | 0.167 | 0.218 | 0.269 | 0.735 |
| LnCSR | 20253 | 3.056 | 0.608 | 0.412 | 2.848 | 3.096 | 3.301 | 4.300 |
| Fiy | 20253 | 0.042 | 0.107 | -0.100 | 0 | 0.010 | 0.037 | 0.720 |
| Oiy | 20253 | 0.060 | 0.068 | -0.125 | 0.020 | 0.052 | 0.091 | 0.301 |
| Size | 20253 | 21.98 | 1.150 | 19.97 | 21.11 | 21.82 | 22.66 | 25.39 |
| MB | 20253 | 0.600 | 0.230 | 0.135 | 0.425 | 0.598 | 0.772 | 1.125 |
| Lev | 20253 | 0.391 | 0.195 | 0.046 | 0.232 | 0.384 | 0.539 | 0.825 |
| ROA | 20253 | 6.923 | 5.720 | -12.23 | 3.550 | 6.205 | 9.750 | 25.60 |
| Mfr | 20253 | 9.166 | 6.383 | 0.978 | 4.816 | 7.809 | 11.64 | 36.33 |
| Fcr | 20253 | 1.163 | 3.013 | -6.830 | -0.225 | 0.671 | 2.036 | 15.48 |
| Grow | 20253 | 15.89 | 27.51 | -41.81 | 0.388 | 11.95 | 26.55 | 134.4 |
| Tax | 20253 | 0.020 | 0.020 | -0.014 | 0.007 | 0.015 | 0.027 | 0.112 |
| Ind | 20253 | 0.373 | 0.093 | 0.176 | 0.300 | 0.364 | 0.429 | 0.625 |
| Econ | 20253 | 0.352 | 0.148 | 0.090 | 0.235 | 0.334 | 0.451 | 0.745 |
| Eres | 20253 | 0.242 | 0.132 | 0.023 | 0.136 | 0.233 | 0.337 | 0.558 |
| Dual | 20253 | 0.279 | 0.449 | 0 | 0 | 0 | 1 | 1 |
| Is | 20253 | 0.401 | 0.252 | 0.001 | 0.167 | 0.417 | 0.612 | 0.887 |
| Ms | 20253 | 0.161 | 0.215 | 0 | 0 | 0.0150 | 0.320 | 0.701 |
| SOE | 20253 | 0.334 | 0.472 | 0 | 0 | 0 | 1 | 1 |
| Cash | 20253 | 0.182 | 0.145 | 0.013 | 0.080 | 0.137 | 0.239 | 0.695 |

0.001, indicating that the highest proportion of financial assets in the sample enterprises is 61.8% and the lowest is 1%. The absolute scale of financialization (CF2) has a minimum value of 14.23 and a maximum of 23.49, which indicates that the samples are extremely different in the degree of financialization. The minimum of CSR score is -0.029, the median is 0.218, the maximum is 0.735, and the average value is 0.240, which indicates that the lowest CSR score in the samples is -2.9, the highest is 73.5, and the average score is 24. Thus, CSR in China is just getting started, and the performance of CSR is significantly different. Distribution of control variables are basically consistent with available literature [19, 54].

## Regression results and analysis

   **(1) Basic regression analysis.**   Table 3 reports the regression results for the test of Hypothesis 1. In the first two columns, we examine the impact of CSR on the relative scale of CF. Column (1) does not add control variables, but only controls the fixed effect of industry, province, and year. The results show that the coefficient of CSR is 0.044, which is significant at 1%. The control variables shown above is added to column (2) of the model, and the result shows that the coefficient of CSR is 0.016, which is still significant at the level of 1%. From an economic point of view, the coefficient of CSR in column (2) means that for every increase in the CSR score, the proportion of financial assets increases by 0.016%. The regression results of the first two columns show that after controlling other factors that affect CF, CSR has a significant role in promoting the allocation of financial assets, and the test results support H1b.

   In the last two columns in Table 3, similar to the first two, we examine the impact of CSR on the absolute scale of CF. Compared with column (3), the regression model in column (4) adds control variables. These two regression results show that the coefficients of CSR are 2.114

Table 3. Impact of corporate social responsibility on corporate financialization.

| | (1) | (2) | (3) | (4) |
|---|---|---|---|---|
| | CF1 | CF1 | CF2 | CF2 |
| CSR | 0.044*** | 0.016*** | 2.114*** | 0.223*** |
| | (7.4576) | (2.6667) | (26.2935) | (3.7925) |
| Controls | N | Y | N | Y |
| Industry | Y | Y | Y | Y |
| Province | Y | Y | Y | Y |
| Year | Y | Y | Y | Y |
| Cons | 0.109*** | 0.043* | 17.700*** | -4.622*** |
| | (12.1111) | (1.8859) | (143.9024) | (-20.7264) |
| Obs | 20253 | 20253 | 20253 | 20253 |
| R-squared | 0.130 | 0.256 | 0.253 | 0.673 |

Note:

*, **, *** are significant at 10%, 5% and 1%, respectively; The value of t is in parentheses. The following tables are the same.

and 0.223 respectively, and both are statistically significant at the level of 1%. From an economic point of view, the coefficient of CSR in column (4) means that the absolute scale of CF increases by 0.223% for every increase in the CSR score. This result also supports the research hypothesis H1b and proves that CSR has "financialization effect".

The above regression results show that CSR mainly plays the role of "reputation insurance" in China. Enterprises undertake CSR not to maximize the stakeholders' value, but to white-wash its short-term investment behavior in pursuit of capital interests. As the return on financial investment is higher than that of industrial investment, enterprises will actively reduce their investment in productive assets and allocate financial assets instead. The profit-seeking color of CF is getting worse [13, 17]. Although the degree to which listed companies fulfill CSR has increased, the management takes still CSR as a strategy to cover up their short-sighted investment behavior.

**(2) A comparative analysis of ownership nature.** The grouping regression results are shown in Table 4. Columns (2) and (4) show that in the sample of SOEs, the impact coefficient of CSR on CF is 0.004 and 0.036 respectively, but it is not statistically significant. Columns (6) and (8) show that in the sample of non-SOEs, the impact coefficient of CSR on CF is 0.039

Table 4. Sample regression of state-owned and non-state-owned enterprises.

| | SOEs | | | | Non-SOEs | | | |
|---|---|---|---|---|---|---|---|---|
| | (1) | (2) | (3) | (4) | (5) | (6) | (7) | (8) |
| | CF1 | CF1 | CF2 | CF2 | CF1 | CF1 | CF2 | CF2 |
| CSR | 0.005 | 0.004 | 1.670*** | 0.036 | 0.067*** | 0.039*** | 1.651*** | 0.373*** |
| | (0.5681) | (0.4494) | (13.6885) | (0.4118) | (8.1707) | (4.6987) | (15.4299) | (4.5710) |
| Controls | N | Y | N | Y | N | Y | N | Y |
| Industry | Y | Y | Y | Y | Y | Y | Y | Y |
| Province | Y | Y | Y | Y | Y | Y | Y | Y |
| Year | Y | Y | Y | Y | Y | Y | Y | Y |
| Cons | 0.156*** | -0.022 | 18.434*** | -4.321*** | 0.071*** | -0.029 | 17.096*** | -5.708*** |
| | (11.4705) | (0.6267) | (98.0531) | (-12.5246) | (5.6800) | (-0.9354) | (104.2439) | (-18.7147) |
| Obs | 6689 | 6689 | 6689 | 6689 | 14000 | 14000 | 14000 | 14000 |
| R-squared | 0.206 | 0.317 | 0.271 | 0.685 | 0.136 | 0.252 | 0.275 | 0.645 |

**Table 5. Sample regression of high ownership concentration and low ownership concentration.**

| | High equity concentration | | | | Low equity concentration | | | |
|---|---|---|---|---|---|---|---|---|
| | (1) | (2) | (3) | (4) | (5) | (6) | (7) | (8) |
| | CF1 | CF1 | CF2 | CF2 | CF1 | CF1 | CF2 | CF2 |
| CSR | 0.049*** | 0.013 | 2.420*** | 0.245*** | 0.043*** | 0.026*** | 1.656*** | 0.270*** |
| | (6.2025) | (1.6250) | (20.6837) | (2.8225) | (4.6739) | (2.7956) | (14.4000) | (3.2335) |
| Controls | N | Y | N | Y | N | Y | N | Y |
| Industry | Y | Y | Y | Y | Y | Y | Y | Y |
| Province | Y | Y | Y | Y | Y | Y | Y | Y |
| Year | Y | Y | Y | Y | Y | Y | Y | Y |
| Cons | 0.094*** | 0.054* | 17.539*** | -4.639*** | 0.130*** | 0.006 | 17.964*** | -4.944*** |
| | (7.8333) | (1.8493) | (97.9832) | (-14.5880) | (9.0277) | (0.1608) | (98.7032) | (-14.7142) |
| Obs | 9997 | 9997 | 9997 | 9997 | 10078 | 10078 | 10078 | 10078 |
| R-squared | 0.168 | 0.287 | 0.296 | 0.680 | 0.120 | 0.235 | 0.226 | 0.656 |

and 0.373 respectively, and both have statistical significance at the level of 1%. The results show that, compared with SOEs, social responsibility of non-SOEs has a significant positive impact on CF, which supports the research hypothesis H2b. For non-SOEs, due to the lack of effective external supervision, CSR has brought a certain amount of external resources, but the influx of external resources has not promoted the investment of entities, but increased the allocation of financial assets of enterprises, causing enterprises to shift "from the real to the virtual economy"; For the SOEs subject to administrative restrictions, their investment decision-making pays more attention to the long-term effect, and the motivation of unethical operation such as over-financialization is relatively weak.

**(3) A comparative analysis of ownership concentration.** To test the influence of internal corporate governance on the relationship between CSR and CF, we take the median of sample corporate ownership concentration as the critical point and divides the sample into high ownership concentration group and low ownership concentration group for grouping regression.

The results are shown in Table 5. Columns (2) and (4) show that for enterprises with high ownership concentration, the impact coefficient of CSR on CF is 0.013 and 0.245 respectively. The former is not statistically significant, while the latter is statistically significant at 1%. Columns (6) and (8) show that for enterprises with dispersed equity, the impact coefficient of CSR on CF is 0.026 and 0.270 respectively, and both are statistically significant at 1%. The results show that compared with the enterprises with concentrated equity, the social responsibility of the enterprises with relatively dispersed equity has a greater and more significant positive impact on CF, which supports the research hypothesis H2b. It shows that the relationship between CSR and CF is negatively regulated when the "governance effect of large shareholders" exists, that is, with the increase of ownership concentration, the positive impact of CSR on CF is weakened. This result shows that the majority shareholder can often play a strong supervisory role in enterprises with high concentration of equity. This conclusion shows that agency problem exists behind the phenomenon of CF.

**(4) A comparative analysis of CSR performance.** There may be differences in the degree of influence of the level of CSR on financialization, because the median of CSR score is the critical point in this paper. The sample is divided into low and high CSR group for grouping regression. The regression results are shown in Table 6. Compared with the low CSR group, the positive relationship between CSR and CF in the high CSR group is more significant, which also confirms the research assumptions H1b and H4b again, that is, CSR has a positive impact on CF. Therefore, under the cover of the good image created by CSR, the agency

**Table 6. Regression of high CSR and low CSR samples.**

| | High CSR groups | | | | Low CSR groups | | | |
|---|---|---|---|---|---|---|---|---|
| | (1) | (2) | (3) | (4) | (5) | (6) | (7) | (8) |
| | CF1 | CF1 | CF2 | CF2 | CF1 | CF1 | CF2 | CF2 |
| CSR | 0.036*** (3.7894) | 0.004 (0.4444) | 2.987*** (23.7063) | 0.145* (1.6628) | 0.056*** (3.3532) | 0.001 (0.0606) | 1.139*** (4.8059) | 0.091 (0.5321) |
| Controls | N | Y | N | Y | N | Y | N | Y |
| Industry | Y | Y | Y | Y | Y | Y | Y | Y |
| Province | Y | Y | Y | Y | Y | Y | Y | Y |
| Year | Y | Y | Y | Y | Y | Y | Y | Y |
| Cons | 0.082*** (4.5810) | 0.085*** (2.5679) | 17.065*** (71.7016) | -4.339*** (-13.4751) | 0.151*** (13.7272) | 0.003 (0.0906) | 18.248*** (116.9743) | -5.060*** (-14.7953) |
| Obs | 10332 | 10332 | 10332 | 10332 | 9922 | 9922 | 9922 | 9922 |
| R-squared | 0.150 | 0.312 | 0.276 | 0.688 | 0.128 | 0.232 | 0.242 | 0.645 |

conflict between managers and shareholders intensifies. Managers may abandon long-term industrial investment and pursue short-term economic benefits through investment in financial assets, which leads to an increase in the proportion of corporate financial assets held.

## Robustness test

**(1) Sensitivity test of CSR: Test the scores of each sub-item of CSR**. The CSR evaluation system of Hexun.com includes Shareholder Responsibility (CSR_sta), Employee Responsibility (CSR_emp), Consumer Responsibility (CSR_con), Environmental Responsibility (CSR_env) and Social Responsibility (CSR_soc). Based on this, we investigate the impact of each sub-item of CSR on CF. Table 7 shows the regression results that the estimation coefficients of CSR_sta, CSR_emp and CSR_soc are 0.749, 0.775 and 0.731, respectively, and are all significant at 1%. The estimation coefficient of CSR_con is 0.465, which is significant at the level of 5%. The

**Table 7. Sensitivity test: Score test for each sub-item of CSR.**

| | (1) | (2) | (3) | (4) | (5) |
|---|---|---|---|---|---|
| | CF2 | CF2 | CF2 | CF2 | CF2 |
| CSR_sta | 0.749*** (4.5951) | | | | |
| CSR_emp | | 0.775*** (2.6223) | | | |
| CSR_con | | | 0.465** (2.4616) | | |
| CSR_env | | | | -0.130 (0.7403) | |
| CSR_soc | | | | | 0.731*** (3.7013) |
| Industry | Y | Y | Y | Y | Y |
| Province | Y | Y | Y | Y | Y |
| Year | Y | Y | Y | Y | Y |
| Controls | Y | Y | Y | Y | Y |
| Cons | -4.681*** (21.0855) | -4.633*** (-20.5911) | -4.668*** (-20.9327) | -4.764*** (-21.2678) | -4.746*** (-21.4751) |
| Obs | 20253 | 20253 | 20253 | 20253 | 20253 |
| R-squared | 0.673 | 0.673 | 0.673 | 0.673 | 0.673 |

**Table 8. Endogeneity test: 2SLS regression.**

|  | (1) | (2) | (3) |
|---|---|---|---|
|  | **CSR** | **CF1** | **CF2** |
| **CSR** |  | 4.056*** | 0.234*** |
|  |  | (23.3103) | (16.0273) |
| **Ind_CSR** | 0.366*** |  |  |
|  | (6.2139) |  |  |
| **Pro_CSR** | 0.750*** |  |  |
|  | (13.2042) |  |  |
| **Controls** | control | control | control |
| **Cons** | -0.794*** | -8.060*** | -0.166*** |
|  | (18.3371) | (32.2400) | (7.2807) |
| **Obs** | 20253 | 20253 | 20253 |
| **R-squared** | 0.319 | 0.542 | 0.0940 |

coefficient of CSR_env is -0.13, which is not statistically significant. This finding is in line with the expectation of this paper, that is, the ratings of shareholder's responsibility, consumer's responsibility and social responsibility have significant promotion effect on CF, while environmental responsibility has certain inhibition effect on CF. This shows that the above three aspects of social responsibility can help enterprises to obtain more resources, and thus promote CF; The resource effect of environmental responsibility is not obvious.

(2) **Endogenous test: 2SLS regression**. This paper mainly studies the impact of CSR on CF, but there is also the possibility of reverse causality, i.e., investment in financial assets will affect CSR decisions. In order to solve the possible endogenous problems, this paper refers to the methods of Quan et al. (2015) and Song (2017) and uses the instrumental variable method to test the robustness of the main conclusions. Specifically, the average level of corporate social responsibility (Ind_CSR) of the same industry in the same year and the average level of corporate social responsibility (Pro_CSR) of the same province in the same year are selected as the tool variables of corporate social responsibility (CSR) for two-stage IV estimation. The regression results are shown in Table 8. The regression results of the first stage show that the coefficients of instrument variables Ind_CSR and Pro_CSR are significantly positive at the level of 1%, with fitting values of 0.366 and 0.750 respectively. This reflects that the industry and provincial average CSR has a significant positive impact on individual corporate social responsibility, that is, corporate social responsibility is contagious. This conclusion is consistent with the existing literature, which also shows that the model has a good explanatory power [54]. The second stage regression results show that the chi-square value of Wald exogeneity test is significant at the level of 1%, indicating that the instrumental variables meet the exogeneity requirements. More importantly, the coefficient of CSR is significantly positive at the level of 1%, indicating that after the use of instrumental variables to mitigate endogenous, CSR is still significantly positively correlated with the level of CF, which is consistent with the previous research findings.

(3) **Other robustness tests**. (a) Lagging CF for one period and estimating the impact of CSR on future corporate financialization. The results are shown in Table 9. The regression coefficients of CSR in columns (2) and (4) are 2.114 and 0.223 respectively, and both are significantly positive at the level of 1%. This conclusion verifies the hypothesis in this paper that CSR has a significant role in promoting CF.

(b) Replace the CSR measurement indicator: (1) Use the natural logarithm (lnCSR) of the Hexun.com CSR score as the explanatory variable to test the indicator sensitivity. The regression results are shown in Table 10. (2) The robustness test is performed using the CSR data in

**Table 9. Robustness test: FA lags one-stage regression.**

|  | (1) | (2) | (3) | (4) |
|---|---|---|---|---|
|  | CF1 | CF1 | CF2 | CF2 |
| CSR | 0.044*** | 2.114*** | 0.016*** | 0.223*** |
|  | (7.4576) | (26.2935) | (2.6667) | (3.7925) |
| Controls | N | Y | N | Y |
| Industry | Y | Y | Y | Y |
| Province | Y | Y | Y | Y |
| Year | Y | Y | Y | Y |
| Cons | 0.109*** | 17.700*** | 0.043* | -4.622*** |
|  | (12.1111) | (143.9024) | (1.8859) | (-20.7264) |
| Obs | 20253 | 20253 | 20253 | 20253 |
| R-squared | 0.130 | 0.253 | 0.256 | 0.673 |

**Table 10. Robustness test: lnCSR regression.**

|  | (1) | (2) | (3) | (4) |
|---|---|---|---|---|
|  | CF1 | CF1 | CF2 | CF2 |
| CSR | 0.009*** | 0.003* | 0.349*** | 0.031** |
|  | (6.4285) | (2.0000) | (17.7157) | (2.1678) |
| Controls | N | Y | N | Y |
| Industry | Y | Y | Y | Y |
| Province | Y | Y | Y | Y |
| Year | Y | Y | Y | Y |
| Cons | 0.091*** | 0.030 | 17.163*** | -4.783*** |
|  | (9.0099) | (1.3100) | (121.7234) | (-21.1637) |
| Obs | 19605 | 19605 | 19605 | 19605 |
| R-squared | 0.131 | 0.260 | 0.240 | 0.674 |

the Rankins CSR Ratings database. Referring to the practice of Song (2017), in order to avoid the result deviation caused by only disclosing the sample of the CSR report, this paper makes the enterprise disclosing the social responsibility report directly take the value as the Runling social responsibility score, and takes the value as 0 for the enterprise not disclosing CSR report, the resulting variable is recorded as rksCSR, and the above regression is performed again with rksCSR as the explanatory variable, and the regression result is shown in Table 11. It can be

**Table 11. Robustness test: rksCSR regression.**

|  | (1) | (2) | (3) | (4) |
|---|---|---|---|---|
|  | CF1 | CF1 | CF2 | CF2 |
| CSR | 0.001*** | 0.001*** | 0.034*** | 0.005*** |
|  | (5.0000) | (5.0000) | (14.1666) | (2.9411) |
| Controls | N | Y | N | Y |
| Industry | Y | Y | Y | Y |
| Province | Y | Y | Y | Y |
| Year | Y | Y | Y | Y |
| Cons | 0.159*** | -0.092* | 18.185*** | -5.785*** |
|  | (7.7941) | (-1.9368) | (65.8876) | (-13.0000) |
| Obs | 4061 | 4061 | 4061 | 4061 |
| R-squared | 0.203 | 0.335 | 0.302 | 0.719 |

seen that the positive correlation between CSR and CF has always been established, which once again confirms the research conclusion.

## Research conclusions and implications

As a typical transition economy country, China has a strong speculative atmosphere in the market, and its regulatory systems are not fully integrated with mature international markets. At present, the internal and external environment of China's capital market is constantly changing, and the impact of CSR on corporate value is still uncertain. This paper empirically tests the relationship between CSR and CF by using the panel data of non-financial listed companies in China from 2009 to 2019. The research results show that: (1) CSR has a financialization effect, i.e., CSR promotes CF through the "reputation insurance effect", which is manifested as follows: although CSR is beneficial for enterprises to obtain key resources to alleviate financial constraints, it also triggers the "investment substitution" motive to obtain high investment returns; (2) Under circumstance of different ownership nature, the fulfillment of CSR has become a tool to promote the degree of CF in non-SOEs, but it is not obvious in SOEs; (3) In case of diverse ownership concentration, CSR has a significant positive impact on CF for enterprises with low ownership concentration, but it is not obvious for high ownership concentration; (4) According to the CSR scores, high CSR score obviously promotes CF, while low CSR score has little effect on CF.

The research significance of this paper is mainly reflected in the following two aspects. First, it provides a literature supplement for the economic consequences of CSR from the viewpoint of reputation insurance effect. Based on the information effect and reputation insurance effect, this paper examines the influencing mechanism of CSR on CF. In a critical period of CSR convergence with modern management practices, these findings provide empirical evidence for the consequences of CSR in transitional economies. Second, this paper contributes a unique view to identifying motives of CF. Most of the existing literature explores CF based on the financial framework. The marginal contribution of this paper is taking CSR as an important non-financial factor for its identification. It helps to exclude the interference of macro and market factors and objectively dissect the financial investment intentions of management.

The policy recommendations are as follows. First, for developing countries, it is currently critical for the construction of CSR system, and regulators should encourage enterprises to optimize the CSR reports. They should disclose non-financial and important financial information such as investment decisions, which helps investors to identify the real motives of CSR. Second, in response to the principal-agent problem, enterprises should improve internal governance structure, bring into the supervisory function of shareholders' meeting, board of directors and supervisors, and reduce the possibility of short-sighted investment behavior. Third, the government should formulate practical reform policies to prevent and resolve excessive financialization of enterprises. In external regulation, consideration should be given to reducing enterprises' operating costs and increasing the profitability of real business. At the same time, the capital support should be strengthened in the entity industry, for example, high-quality assets should be revitalized through financial model innovation, so that financial capital can really empower the industrial economy.

## Limitations

Firstly, this paper could further study the evolvement course of CF for Chinese enterprises form a broader perspective of globalization. Since CF is a promising development trend in the future, it is of great theoretical and real significance to analyze the impact of CSR on CF in a global view. Secondly, of all the diverse dimensions of CSR, this paper emphasizes on the

contradictory effecting mechanism between shareholder and social responsibility. Research on the interactions of the other dimensions in the future will bring great effects on study of the CF. Thirdly, some other comprehensive factors impacting CF should be included in the future study, such as geographical features, cultural atmosphere, and regional economic features with potential impacts. All these aspects will lead to more research with more optimized and in-depth analysis.

## Author Contributions

**Conceptualization:** Lei Lei, Di Zheng.

**Data curation:** Lei Lei.

**Formal analysis:** Lei Lei, Di Zheng.

**Investigation:** Lei Lei, Di Zheng.

**Methodology:** Lei Lei, Di Zheng, XuDong Chen.

**Resources:** Lei Lei, Di Zheng.

**Supervision:** XuDong Chen.

**Writing – original draft:** Lei Lei, Di Zheng.

**Writing – review & editing:** Lei Lei, Di Zheng, XuDong Chen.

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
