## [Decision Letter · Decision Letter 0]

19 Apr 2022

PONE-D-22-01138Corporate Social Responsibility and Corporate Financialization — Based on Information Effect and Reputation Insurance EffectPLOS ONE

Dear Dr. Lei,

Thank you for submitting your manuscript to PLOS ONE. After careful consideration, we feel that it has merit but does not fully meet PLOS ONE’s publication criteria as it currently stands. Therefore, we invite you to submit a revised version of the manuscript that addresses the points raised during the review process.

I have just received the feedback from two expert reviewers in your field and both of them are very positive about your research. Sorry about the delay but I feel the wait will have been worthwhile for you. Note that both reviewers have made suggestions about how to improve the clarity and completeness of your paper, so please make sure to follow their advice when preparin a revised version of your manuscript. I will re-review your new version myself to speed up the process.

We look forward to receiving your revised manuscript.

Kind regards,

Iván Barreda-Tarrazona, PhD

Academic Editor

PLOS ONE

Journal Requirements:

Reviewers' comments:

Reviewer's Responses to Questions

**Comments to the Author**

1. Is the manuscript technically sound, and do the data support the conclusions?

Reviewer #1: Yes

Reviewer #2: Yes

2. Has the statistical analysis been performed appropriately and rigorously? 

Reviewer #1: Yes

Reviewer #2: Yes

3. Have the authors made all data underlying the findings in their manuscript fully available?

Reviewer #1: Yes

Reviewer #2: Yes

4. Is the manuscript presented in an intelligible fashion and written in standard English?

Reviewer #1: Yes

Reviewer #2: Yes

5. Review Comments to the Author

Reviewer #1: Thank you for the opportunity of reading and reviewing your manuscript. The paper is well written with a rigorous reserach and addresses a topic with high importance. I suggest amending the paper in its final section, e.g. there is a very short reference to the meaning of your findings. Please consider discussing the practical and policy implications of your findings, their utility in wider context and also refer to limitations and further studies. Good luck!

Reviewer #2: This paper is very interesting to explore the relationship between CSR and financialization, and considers the information effect and reputation insurance effect. The research design is good, and the empirical analysis can support the research findings. Some improvements should be considered in this paper:

1. In the section of introduction, the authors should establish the importance of this research topic from a global perspective.

2. In the section of 3.2, the variable of financialization should be corporate financialization, just like your title.

3. In the robustness test, the measurement of CSR can use the other data source, such Hexun.

6. PLOS authors have the option to publish the peer review history of their article (what does this mean?). If published, this will include your full peer review and any attached files.

Reviewer #1: No

Reviewer #2: No

---

## [Author Response · Author response to Decision Letter 0]

17 Jun 2022

Dear Editor and Reviewers:

Thank you for giving us the opportunity to submit a revised draft of the manuscript “Corporate Social Responsibility and Corporate Financialization — Based on Information Effect and Reputation Insurance Effect” for publication in PLOS ONE. We appreciate the time and effort that you and the reviewers dedicated to providing feedback on our manuscript and are grateful for the insightful comments on and valuable improvements to our paper.

We have incorporated most of the suggestions made by the reviewers. Those changes are highlighted within the manuscript. Please see below, in red, for a point-by-point response to the reviewers’ comments and concerns. All page numbers refer to the revised manuscript with tracked changes.

Response to Reviewer #1:

We are extremely grateful to the reviewer for pointing out this problem. As suggested by the reviewer, we have added the meaning of our findings in the “Research Conclusions and Implications” part. Specifically, we discussed the theoretical significance of this paper in Paragraph 2, line 472-482., and the illumination for policymaking in Paragraph 3, line 483-496. Besides, we summarize the limitations of this paper separately and put it in the Limitations part. We illustrate the aspects which should have been taken into consideration and worthy to be exploded in the coming research in line 498-508. 

Response to Reviewer #2:

Thanks for your generous comments. According to your advice, we amended the relevant part in the manuscript. All your questions were answered one-by-one.

1. We believe that the reviewer's comments are highly relevant and help us to see the significance and value of the topic more clearly. Therefore, based on the reviewer's comments, we have added a relevant background introduction on the trend of financialization of the economy (Paragraph 1&2, lines 2-18) and the global CSR development (Paragraph 3, lines 30-37) in the Introduction part, which can further highlight the importance of the research topic from a global perspective.

2. Based on the reviewers’ comments, we have corrected the naming of the financialization variable in the Measurement of Variables part (Line 266), which is consistent with the title and context of this paper.

3. In fact, in the basic regression model of this paper, the CSR score index of Hexun.com is selected to measure the CSR performance. Meanwhile, in the robustness test, the CSR report disclosure of Runling Global is used for further tests. In China, there are only two authoritative and publicly available CSR databases, Hexun.com and Runling Global. Currently, there are no other reliable sources of CSR data that can be used for a variety of robustness tests.

Once again, thank you and all the reviewers for the kind advice and vigorous help. If you have any questions, please contact us without hesitate.

Yours sincerely,

Lei Lei

---

## [Editor Report · Decision Letter 1]

4 Jul 2022

Corporate social responsibility and corporate financialization — based on information effect and reputation insurance effect

PONE-D-22-01138R1

Dear Dr. Lei,

We’re pleased to inform you that your manuscript has been judged scientifically suitable for publication and will be formally accepted for publication once it meets all outstanding technical requirements.

Kind regards,

Iván Barreda-Tarrazona, PhD

Academic Editor

PLOS ONE

Additional Editor Comments (optional): You have successfully addressed all the topics raised by the reviewers. Congratulations.
---

## [Editor Report · Acceptance letter]

7 Jul 2022

PONE-D-22-01138R1 

Corporate social responsibility and corporate financialization — based on information effect and reputation insurance effect 

Dear Dr. Lei:

I'm pleased to inform you that your manuscript has been deemed suitable for publication in PLOS ONE. Congratulations! Your manuscript is now with our production department. 

Kind regards, 

on behalf of

Dr. Iván Barreda-Tarrazona 

Academic Editor

PLOS ONE